# Attribute-driven Disentangled Representation Learning for Multimodal Recommendation

## ABSTRACT

Recommendation algorithms predict user preferences by correlating user and item representations derived from historical interaction patterns. In pursuit of enhanced performance, many methods focus on learning robust and independent representations by disentangling the intricate factors within interaction data across various modalities in an unsupervised manner. However, such an approach obfuscates the discernment of how specific factors (e.g., *category* or *brand*) influence the outcomes, making it challenging to regulate their effects. In response to this challenge, we introduce a novel method called Attribute-Driven Disentangled Representation Learning (short for AD-DRL), which explicitly incorporates attributes from different modalities into the disentangled representation learning process. By assigning a specific attribute to each factor in multimodal features, AD-DRL can disentangle the factors at both attribute and attribute-value levels. To obtain robust and independent representations for each factor associated with a specific attribute, we first disentangle the representations of features both within and across different modalities. Moreover, we further enhance the robustness of the representations by fusing the multimodal features of the same factor. Empirical evaluations conducted on three public real-world datasets substantiate the effectiveness of AD-DRL, as well as its interpretability and controllability.

## CCS CONCEPTS

• **Information systems** → **Personalization**; **Recommender systems**; **Collaborative filtering**.

## KEYWORDS

Multimodal, Attribute, Disentangled Representation, Recommendation

## 1 INTRODUCTION

Recommender systems (RS) are integral to a myriad of online platforms, spanning E-commerce to advertising, facilitating users in pinpointing items aligned with their preferences. Given their pivotal role, numerous efforts have been dedicated to developing advanced recommendation models to improve their performance. Among them, Collaborative Filtering (CF) based models [9, 10, 12, 26, 27, 34, 35] have achieved great success by exploiting the user-item interaction data to learn user and item representation. However,

*ACM MM, 2024, Melbourne, Australia*
© 2024 Copyright held by the owner/author(s). Publication rights licensed to ACM.
ACM ISBN 978-x-xxxx-xxxx-x/YY/MM
https://doi.org/10.1145/nnnnnnn.nnnnnnn

**Unpublished working draft. Not for distribution.**

these models can easily encounter the sparsity problem in practical scenarios due to their exclusive dependence on interaction data. To alleviate the data sparsity problem in recommendation, side information (e.g., attributes, user reviews and item images), which contains rich information associated with users or items, is often used to enhance the representation learning of users and items [17, 19, 37–39, 41].

It is well recognized that the entangled representations of users and items are infeasible to directly capture fine-grained user preferences across diverse factors, thereby constraining both the efficacy and interpretability of recommender systems [21, 35]. In recent years, the study of disentangled representation learning has garnered significant attention in diverse fields, notably in computer vision [11, 13, 20], due to its capability to identify and disentangle the underlying factors behind data. Empirical evidence suggests that disentangled representations exhibit enhanced robustness, particularly in complex application contexts. Hence, many recommendation methods adopt disentangled representation learning techniques to learn robust and independent representations, ultimately enhancing recommendation performance [17, 21, 23, 31, 35]. For example, Ma et al. [21] employed disentangled representations to capture user preferences regarding different concepts associated with user intentions. Wang et al. [35] introduced a GCN-based model that produces disentangled representations by modelling a distribution over intents for each user-item interaction, exploring the diversity of user intentions on adopting items. However, these methods only concentrate on disentangling the user and item representations based on their ID embeddings. To exploit the difference between the factors behind data of various modalities, Liu et al. [17] estimated the users' attention weight to underlying factors of different modalities, utilizing a sophisticated attention-driven module.

Despite the considerable advancements brought about by disentangling techniques in recommender systems, existing studies often disentangle both users and items into latent factors. This methodology inherently constrains the model's interpretability and controllability. For example, consider using disentangled representations to elucidate the diverse factors, such as *style*, *brand*, *popularity*, and *price*, that shape user preferences in dress selection. The inherent abstraction of latent factors from current disentangling methods makes it difficult to pinpoint which factor represents each specific dress attribute. In particular, one factor might be loosely related to a combination of *brand* and *price*, while another factor might represent a mixture of *style* and *popularity*. This ambiguity in the latent factors limits the interpretability of the recommendation system, making it difficult to understand why a particular item was recommended to a user. Moreover, this lack of clarity also affects the controllability of the recommendation system. Suppose a user wants to receive recommendations specifically for dresses without considering her preferences for *price* and *popularity*. It would be challenging to adjust the recommender system to focus on those

specific preferences, as the latent factors are not clearly tied to these attributes.

Item attributes manifested in various modalities can enrich the recommendation system by offering diverse and complementary information. For example, the textual data might explicitly mention the *brand* or *price*, while visual data can reveal visual attributes, such as the *category* of items. Additionally, the *popularity* of items can be derived from the statistical information of interaction data. In fact, attributes represent specific, meaningful properties or characteristics of items. Consequently, using attributes to guide the disentanglement process is a promising way to improve the interpretability and controllability of conventional multimodal recommendation methods. In this paper, we propose an **A**ttribute-**D**riven **D**isentangled **R**epresentation **L**earning method (AD-DRL for short), which disentangles factors in user and item representations across various modalities at different levels of attribute granularity. To obtain robust and independent representations for each factor associated with a specific attribute, we first disentangle the representations of features within and across different modalities. This process is guided by high-level attributes (e.g., *category* and *popularity*), which help reveal the underlying relationships between factors. Following this, we further enhance the robustness of the representations by fusing the multimodal features of the same factor. This step involves exploiting the relationships among representations of the same factor by leveraging low-level attributes (e.g., *category* attribute values for a clothing dataset, such as *jeans*, *jackets*, and *dresses*), resulting in finer-grained and more comprehensive disentangled representations. To validate the effectiveness of our method, we conduct extensive experiments and ablation studies on three real-world datasets. Experimental results demonstrate the superiority of our method AD-DRL compared to existing methods and showcase its capability in terms of interpretability and controllability.

In summary, the contributions of this paper are threefold:

- In this paper, we highlight the limitations of traditional disentangled representation learning in multimodal recommendation systems. To overcome these shortcomings, we introduce AD-DRL, which improves interpretability and controllability by employing attributes to disentangle factors in user and item representations.
- To achieve robust and independent representations, we assign a specific attribute to each factor in multimodal features and disentangle factors at both the attribute and attribute-value levels.
- We conduct extensive experiments on three real-world datasets to validate the effectiveness of our method. The experimental analysis demonstrates the interpretability and controllability of our model.

## 2 RELATED WORK

### 2.1 Multimodal Collaborative Filtering

Traditional Collaborative Filtering (CF) [4, 27, 28] methods primarily rely on user-item interactions to learn representations of users and items. Consequently, the recommendation performance is negatively impacted when encountering users and items with limited interactions. To mitigate the challenges posed by data sparsity [14, 18, 22, 29], recent research has incorporated multimodal

information into recommendation systems. The multimodal features, such as reviews and images, provide valuable information for user preference and item characteristics, which can supplement historical user-item interactions and thus enhance recommendation performance [8, 17, 19, 37, 38, 41].

Previous studies integrate multimodal information into the matrix factorization-based method in a straightforward manner. For instance, VBPR [8] directly concatenates the visual features learned from item images with collaborative features as the joint item representation and feeds it into the MF module. With the success of deep learning techniques in modeling complex interaction behaviors [10] and the relation between various multimodal features [24], a lot of deep learning techniques have been introduced into multimodal recommendation [10, 30]. For example, MAML [19] first fuses the item's multimodal features and user features, then feeds it into an attention module to capture users' diverse preferences. VECF [1] constructs a visually explainable collaborative filtering model using a multimodal attention network, facilitating the integrated coupling of diverse feature modalities. More recently, graph convolutional networks (GCNs) [7] have demonstrated their power capability of representation learning for recommendation [9, 34, 36, 43]. Based on the GCN structure, MMGCN [38] constructs a user-item bipartite graph to learn representations of each modality and then fuse the modality representation together as the final representation. GRCN [37] utilizes the rich multimodal content of items to refine the structure of the interaction graph in order to mitigate the effect of false-positive edges on recommendation performance.

### 2.2 Disentangled Representation Learning

Disentangled representation learning, which seeks to identify and separate underlying explanatory factors within data, has garnered significant interest, especially in the field of computer vision [5, 11, 20]. For instance, the beta-VAE method [11] employs a constrained variational framework to learn disentangled representations of fundamental visual concepts. Meanwhile, IPGDN [20] automatically uncovers independent latent factors present in graph data.

Owing to the successful application of disentangled representation learning in various domains, numerous studies have concentrated on learning disentangled representations for users and items in recommendation systems in recent years [17, 21, 23, 31, 33, 35]. Following the previous work in computer vision, the initial attempts take Variational Auto-encoder (VAE) [16] to learn disentangled representations. For example, MacridVAE [21] captured user preferences regarding the different concepts associated with user intentions separately. Beyond the disentangled representations derived from user-item interaction modeling, ADDVAE [31] acquires an additional set of disentangled user representations from textual content and subsequently aligns both sets of representations. For studying the diversity of user intents on adopting the items, DGCF [35] employs a graph disentangled module to iteratively refine the intent-aware interaction graph and factorial representations for recommendations. In order to model the users' various preferences on different factors of each modality, DRML [17] estimates the user's attention weight to underlying factors of different modalities with an attention module. As the learned disentangled representations lack clear meanings, KDR [23] harnesses the power

of knowledge graphs (KGs) to guide the disentangled representation learning process, ensuring that the resulting disentangled representations are associated with semantically meaningful information extracted from the KGs.

Although the existing models achieve performance improvement by disentangled representation learning, they all face the problem that they all disentangle the user and item representation into latent factors without clarifying the semantic meaning of each factor.

## 3 METHOD

### 3.1 Preliminaries

*3.1.1 Problem Setting.* Given a set of users $\mathcal{U} \in \{u\}$ and items $\mathcal{I} \in \{i\}$, we utilize two types of information to learn user and item representations: (1) **A user-item interaction matrix $R$**, where each entry $r_{ui} \in R$ represents the implicit feedback (*e.g.*, clicks, likes or purchases) of user $u$ to item $i$. $r_{ui} = 1$ indicates that user $u$ has interacted with item $i$ and $r_{ui} = 0$ indicates that there is no interaction between user $u$ and item $i$ in the observed data; and (2) **Multimodal features**, which mainly consist of two types of features associated with items, namely *reviews* and *images*. By utilizing user-item interactions and multimodal features, we aim to learn robust representations of users and items, thereby predicting a user's preference for items they have not yet interacted with.

*3.1.2 Notations.* Similar to previous work [17, 37], each user $u$ and item $i$ are assigned with a unique ID and respectively represented by a vector $\boldsymbol{v}_u \in \mathbb{R}^d$ and $\boldsymbol{v}_i \in \mathbb{R}^d$, which are randomly initialized in our model. For the review and image information, we use the BERT [2] and the ViT [3] to extract the raw textual features $\boldsymbol{e}_t \in \mathbb{R}^{d_0}$ and visual features $\boldsymbol{e}_v \in \mathbb{R}^{d_0}$, respectively. To tailor the recommendation-oriented features, we adopt two non-linear transformations to cast $\boldsymbol{e}_t$ and $\boldsymbol{e}_v$ into the same feature space:

$$\begin{aligned} \boldsymbol{v}_t &= \sigma(\boldsymbol{W}_t \boldsymbol{e}_t + \boldsymbol{b}_t), \\ \boldsymbol{v}_v &= \sigma(\boldsymbol{W}_v \boldsymbol{e}_v + \boldsymbol{b}_v), \end{aligned} \quad (1)$$

where $\boldsymbol{W}_t, \boldsymbol{W}_v \in \mathbb{R}^{d \times d_0}$ and $\boldsymbol{b}_t, \boldsymbol{b}_v \in \mathbb{R}^d$ denote the weight matrices and bias vectors for textual and visual modality, respectively. $\sigma(\cdot)$ is the activation function.

Following the disentangled representation learning process in previous work [17, 35], we first split the feature vector into $K$ chunks. Each chunk corresponds to a specific item attribute, such as *price*, *brand*, etc. For simplicity, we equally split the representation of each modality into $K$ continuous chunks. Take item ID embedding as an example:

$$\boldsymbol{v}_i = (\boldsymbol{v}_i^1, \boldsymbol{v}_i^2, \cdots, \boldsymbol{v}_i^K), \quad (2)$$

where $\boldsymbol{v}_i^k \in \mathbb{R}^{\frac{d}{K}}$ is the item ID embedding corresponding to the $k$-th attribute. Analogously, $\boldsymbol{v}_t = (\boldsymbol{v}_t^1, \boldsymbol{v}_t^2, \cdots, \boldsymbol{v}_t^K)$, $\boldsymbol{v}_v = (\boldsymbol{v}_v^1, \boldsymbol{v}_v^2, \cdots, \boldsymbol{v}_v^K)$ and $\boldsymbol{v}_u = (\boldsymbol{v}_u^1, \boldsymbol{v}_u^2, \cdots, \boldsymbol{v}_u^K)$ are defined as the embeddings of textual feature, visual feature and user ID embedding, respectively.

*3.1.3 Intuition of Attribute-driven Disentanglement.* Our work not only aims to recommend accurate items to users but also advocates for attribute-driven disentanglement in multimodal recommender systems to enhance the interpretability and controllability of recommendations. Specifically, unlike the existing method that disentangles the latent factors in an unsupervised manner, we leverage

the semantic labels of item attributes to learn attribute-specific subspaces for each attribute to obtain disentangled representations. In this paper, the vector of each modality is composed of $K$ chunks, with the assumption that each chunk is associated with an attribute (such as *price* or *brand*). The achievement of attribute-driven disentangling requires two necessary conditions. Firstly, each chunk should have a clear attribute reference, and chunks associated with different attributes should be distinguishable. For example, chunk $j$ is associated with the *price*, while chunk $k$ represents the vector of the *brand*. Secondly, each chunk of a specific attribute should accurately capture the specific attribute value. For example, chunk $j$ should reflect the price level (i.e., expensive or cheap) of the product.

### 3.2 Attribute-driven Disentangled Representation Learning

In this section, we elaborate on our proposed model, termed AD-DRL, an acronym for *Attribute-Driven Disentangled Representation Learning model*. We aim to improve the interpretability and controllability of recommendation models by assigning a specific attribute to each factor in multimodal features. Specifically, AD-DRL comprises two disentangling modules: high-level and low-level attribute-driven disentangled representation learning. In the *high-level attribute-driven disentangled representation learning* module, we exploit the difference between attribute factors within the same modality feature and the consistency of the same attribute factor across different modalities. In contrast, the *low-level attribute-driven disentangled representation learning* module leverages the intrinsic relationships between items sharing the same attribute value.

*3.2.1 High-Level Attribute-driven Disentangled Representation Learning.* To obtain robust and independent representations for each attribute factor in multimodal features, AD-DRL enables disentangling within each modality feature and ensures consistency of representations across modalities. Next, we detail the intra-modality disentanglement and inter-modality disentanglement.

**Intra-Modality Disentanglement.** After obtaining the feature vectors of each modality, we split these vectors into several chunks. Nevertheless, different attribute factors are still entangled within the chunks. In other words, a chunk may contain both *price*- and *brand*-related features. Therefore, in order to disentangle attribute factors in each modality feature, we employ an attribute classifier to encourage each chunk to predict the corresponding attribute, as shown in Figure 1a. Taking the $k$-th chunk $\boldsymbol{v}_t^k$ of item textual embedding as an example,

$$\begin{cases} z^k &= \boldsymbol{W}_{intra,t} \boldsymbol{v}_t^k + \boldsymbol{b}_{intra,t}, \\ l_{intra,t}^k &= -\sum_{n=1}^{K} \tilde{z}_n^k \log \frac{\exp z_n^k}{\sum_m \exp z_m^k} \end{cases} \quad (3)$$

where $\boldsymbol{W}_{intra,t} \in \mathbb{R}^{K \times \frac{d}{K}}$ and $\boldsymbol{b}_{intra,t} \in \mathbb{R}^K$ are the weight matrix and bias vector of the classifier, respectively. $z^k$ is the predicted logits and $z_n^k \in z^k$. $\tilde{z}_n^k$ denotes the ground truth (attribute label) of chunk $\boldsymbol{v}_t^k$. For all the chunks in textual embedding $\boldsymbol{v}_t$, we have the following loss for all attribute factors in textual features:

$$l_{intra,t} = \sum_{k=1}^{K} l_{intra,t}^k. \quad (4)$$

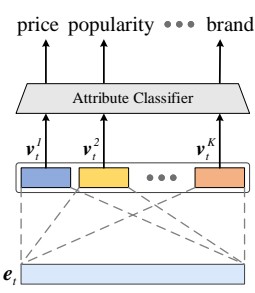

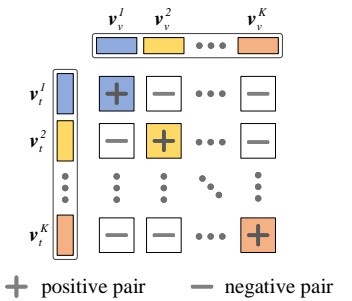

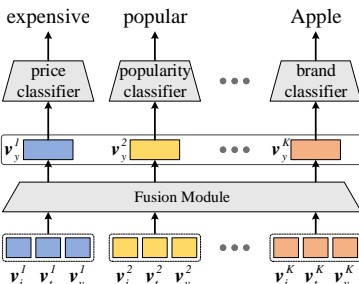

(a) High-level attribute-driven disentangled representation learning (Intra)

(b) High-level attribute-driven disentangled representation learning (Inter)

(c) Low-level attribute-driven disentangled representation learning

**Figure 1: High-level and low-level attribute-driven disentangled representation learning module of our proposed AD-DRL. To disentangle attribute factors in multimodal features, (a) Intra-modality disentanglement module exploits the difference between attribute factors (e.g., *price, brand, category* and *popularity*) within the same modality feature, (b) Inter-modality disentanglement module utilizes the consistency of the same attribute factor in different modality features, and (c) low-level disentangled representation learning module leverages the intrinsic relationships between items sharing the same attribute value (e.g., the popularity value of different levels: *Super Popular, Popular, Moderate, Emerging,* and *Unknown*).**

Similarly, we can define the losses $l_{intra,u}$, $l_{intra,i}$ and $l_{intra,v}$ to encourage the features of each attribute factor to be concentrated in the corresponding chunk of user ID, item ID and visual feature, respectively. Finally, the total loss for the intra-modality disentanglement is formulated as:

$$l_{intra} = l_{intra,u} + l_{intra,i} + l_{intra,t} + l_{intra,v}. \quad (5)$$

**Inter-Modality Disentanglement.** In addition to separately disentangling each modality, a serious challenge in disentangling representation learning for multi-modal features is handling the inter-relationship between the factors disentangled from multiple modalities. Intuitively, the chunks of the same attribute factor from different modality features should be consistent. For example, for the *brand* attribute, an item should have the same brand information in both the visual and textual features. In other words, chunks that share the same attribute across different modalities should be highly similar, while chunks that represent different attributes across modalities should be dissimilar.

To further achieve robust representation in multimodal recommendations, we disentangle attribute factors in different modality features by ensuring consistency of the same attribute across modalities. Inspired by this, we design a cross-modal contrastive loss as shown in Figure 1b. Specifically, for any two modality representations, such as $v_t$ and $v_v$, we take two chunks of the same attribute factor (i.e., $(v_t^k, v_v^k)$) as positive pairs, and two chunks corresponding to different attributes (i.e., $(v_t^k, v_v^{n \neq k})$, $(v_t^{n \neq k}, v_v^k)$) as negative pairs. After that, the cross-modal contrastive loss between textual and visual features is defined as follows:

$$\begin{cases} l_{t \to v}^k &= -\log \frac{\exp((v_t^k \cdot v_v^k)/\tau)}{\sum_{n=1}^{K} \exp((v_t^k \cdot v_v^n)/\tau)}, \\ l_{v \to t}^k &= -\log \frac{\exp((v_v^k \cdot v_t^k)/\tau)}{\sum_{n=1}^{K} \exp((v_v^k \cdot v_t^n)/\tau)}, \\ l_{t \leftrightarrow v} &= \sum_{k=1}^{K} l_{t \to v}^k + \sum_{k=1}^{K} l_{v \to t}^k, \end{cases} \quad (6)$$

where $\cdot$ represents the dot product and $\tau \in \mathbb{R}^+$ is a scalar temperature parameter. By applying this contrastive learning constraint to any two out of the three modalities, we can achieve disentanglement

and alignment across features of various modalities:

$$l_{inter} = l_{i \leftrightarrow t} + l_{i \leftrightarrow v} + l_{t \leftrightarrow v}. \quad (7)$$

*3.2.2 Low-level Attribute-driven Disentangled Representation Learning.* Each attribute $k$ is associated with a list of possible attribute values $(\tilde{y}_1^k, \tilde{y}_2^k, \cdots, \tilde{y}_{A_k}^k)$, where $A_K$ is the total number of possible values for that attribute. For example, the attribute value of *popularity* can be stratified into five distinct levels: Super Popular, Popular, Moderate, Emerging, and Unknown. The representation disentanglement could benefit from the attribute values by exploiting their relationships of the same factor. Therefore, to learn more robust representations, we encourage disentangled representations based on attributes to predict specific attribute values of the item.

It is obvious that any single modality may not contain sufficient information for a particular attribute. For example, images may more intuitively reflect attributes such as *color* and *brand* of an item, but may not directly reflect attributes such as *price* or *popularity*. Therefore, in order to comprehensively and accurately depict the attribute values of the item, we first integrate the features from all modalities together. To achieve this, we apply a multimodal attention mechanism to measure the emphasis of different modalities on different attributes. Specifically, for the $k$-th attribute, the attention weights assigned to different modalities are estimated by a two-layer neural network:

$$\begin{cases} \hat{a}^k = W_{a2} \tanh(W_{a1}(v_i^k + v_t^k + v_v^k) + b_a), \\ a^k = Softmax(\hat{a}^k), \end{cases} \quad (8)$$

where $W_{a1} \in \mathbb{R}^{3 \times \frac{d}{K}}$ and $W_{a2} \in \mathbb{R}^{3 \times 3}$ are the weight matrices corresponding to the first and second layers of the neural network, respectively. $b_a$ denotes the bias vector and tanh is the activation function. $Softmax$ is adopted to normalize $\hat{a}^k$ to a probability distribution.

Thereafter, we obtain the final representation of the $k$-th attribute factor for the item as follows:

$$v_y^k = a_i^k \cdot v_i^k + a_t^k \cdot v_t^k + a_v^k \cdot v_v^k, \quad (9)$$

where $a_i^k, a_t^k, a_v^k \in \boldsymbol{a}^k$ are the attention weights of item ID embedding, textual feature and visual feature, respectively.

Similar to the disentangling at the intra-modality disentanglement in Section 3.2.1, we aim to achieve disentangled representation learning at the low level through attribute value prediction via a classification layer (as shown in Figure 1c):

$$\begin{cases} \boldsymbol{y}^k & = W_k \boldsymbol{v}_y^k + \boldsymbol{b}_k, \\ l_{low}^k & = -\sum_{i=1}^{A_K} \tilde{y}_i^k \log \frac{\exp y_i^k}{\sum_j \exp y_j^k}, \\ l_{low} & = \sum_{k=1}^{K} l_{low}^k \end{cases} \tag{10}$$

where $W_k \in \mathbb{R}^{K \times \frac{d}{K}}$ and $\boldsymbol{b}_k \in \mathbb{R}^K$ are the weight matrices and bias vector of the classifier, respectively. We supervise the training of each attribute subspace via independent attribute classification tasks defined in the form of a cross-entropy loss.

## 3.3 Preference Prediction and Model Learning

### 3.3.1 Preference Prediction.
So far we have discussed how to obtain attribute-driven disentangled representations $\boldsymbol{v}_u = (\boldsymbol{v}_u^1, \boldsymbol{v}_u^2, \cdots, \boldsymbol{v}_u^K)$ and $\boldsymbol{v}_y = (\boldsymbol{v}_y^1, \boldsymbol{v}_y^2, \cdots, \boldsymbol{v}_y^K)$ for the user $u$ and item $i$, respectively. By assigning each disentangled representation with an attribute as described above, we are able to predict users' preferences at the attribute level. This enhances the interpretability and controllability of our model. Specifically, to estimate a user $u$'s preference for an item $i$, it is crucial to consider her preference for each attribute of the item. To achieve this, we first compute the user's preference score for each attribute of the item, and subsequently, integrate the scores of each attribute to estimate the user's overall preference for the item:

$$\begin{cases} s_{u,i,k} = \sigma(\boldsymbol{v}_u^k \cdot \boldsymbol{v}_y^k), \\ s_{u,i} = \sum_{k=1}^{K} s_{u,i,k}, \end{cases} \tag{11}$$

where $\cdot$ symbolizes the dot product and $\sigma(\cdot)$ denotes the activation function. In this paper, we use a softplus function to ensure the resultant score $s_{u,i,k}$ is positive. $s_{u,i,k}$ denotes the user $u$'s preference score for the attribute $k$ of the item $i$.

### 3.3.2 Training Protocol.
Based on the predicted preference score above, we recommend a list of top-$n$ ranked items that match the target user's preferences. The Bayesian Personalized Ranking (BPR) loss function [27] is employed to optimize the model parameters $\Theta$,

$$\mathcal{L}_{BPR} = \sum_{(u,i_+,i_-) \in \mathcal{D}} -\log \phi(s_{u,i_+} - s_{u,i_-}) + \lambda \|\Theta\|_2^2, \tag{12}$$

where $\lambda$ is the coefficient controlling $L_2$ regularization; $\mathcal{D}$ denotes the training set; $i_+$ and $i_-$ are the observed and unobserved items in the interaction records of user $u$, respectively. Overall, the total loss of AD-DRL is formulated as,

$$\mathcal{L} = \mathcal{L}_{BPR} + \alpha \sum_{(u,i) \in \mathcal{D}} l_{intra} + \beta \sum_{(u,i) \in \mathcal{D}} l_{inter} + \gamma \sum_{(u,i) \in \mathcal{D}} l_{low}, \tag{13}$$

where $\alpha$, $\beta$ and $\gamma$ are the hyperparameters to control the weight of three disentanglement modules.

## 4 EXPERIMENTS

### 4.1 Experimental Setup

#### 4.1.1 Datasets.
We use the widely adopted real-world recommendation dataset, the Amazon review dataset[1] [22], for evaluation in our experiments. Apart from user-item interaction data, this dataset also includes multimodal information (i.e., reviews and images) and various attributes (i.e., price, brand, etc.) of the items on 24 product categories. Three product categories from this dataset are used in the evaluation: Baby, Toys Games and Sports. Following [17], for all datasets, unpopular items and inactive users are filtered out to ensure that all the items and users have at least 5 interaction records. The basic statistics of the three datasets are shown in Table 1.

In our setting, in addition to the user-item interaction data and multimodal information of items, multiple attributes and their associated attribute values of items are required to guide the disentangled representation learning process. Specifically, we adopt four typical attributes (i.e., price, popularity [2], brand and category), and the attribute values for each attribute are compiled based on the metadata provided by the Amazon dataset: for brand and category values, we directly used the values provided by Amazon; for price and popularity, following the methods used by CoHHN [40], we discretize them into five levels according to their numerical values. Table 1 shows the specific statistical information for each attribute in our experiments. It is worth noting that our method can include a wide range of attributes that objectively reflect the character of an item, beyond the four attributes mentioned.

#### 4.1.2 Baselines.
We compared our method with the state-of-the-art methods, including both the CF-based methods (i.e., NeuMF [10], NGCF [34] and DGCF [35]) and Multimodal CF-based methods (JRL [42], MMGCN [38], MAML [19], GRCN [37], DMRL [17] and BM3 [43]). Each type of method includes a disentangled representation learning model (i.e., DGCF and DMRL) for comparison. In addition to the methods mentioned above, we also created a variant of our model, denoted as AD-DRL$_{ID}$, which excludes the utilization of multimodal information, thus facilitating a fair comparison with CF-based models.

#### 4.1.3 Evaluation Metrics and Parameter Settings.
For each dataset, we randomly split the interactions from each user with an $8:2$ ratio to construct training and testing sets. From the training set, 10% of interactions are randomly chosen as a validation set for tuning hyperparameters. We evaluate performance on the top-$n$ recommendation task, aiming to recommend the top-$n$ items, using Recall@$n$ and NDCG@$n$ as metrics for accuracy, with $n$ defaulting to 20.

The Pytorch toolkit [25] is utilized to implement our models. To ensure fairness, all methods are optimized using the Adam optimizer [15], with a default learning rate of 0.0001 and batch size of 1024. We fixed the embedding size of each factor to 32 for AD-DRL and its variants on all datasets. The Xavier initializer [6] is used to initialize the model parameters. The $\alpha$, $\beta$, $\gamma$ and $L_2$ regularization coefficient are searched in $\{1e^{-3}, 5e^{-3}, 1e^{-2}, \cdots, 5e^{+0}, 1e^{+1}\}$. The number of negative examples is searched in $\{2, 4, 8\}$. Besides,

---

[1] http://jmcauley.ucsd.edu/data/amazon.

[2] Popularity is calculated by counting the number of times each item is purchased.

**Table 1: Basic statistics of the three datasets. "#" denotes the number of statistical values.**

| Dataset | #user | #item | #interaction | sparsity | #price | #popularity | #brand | #category |
|---|---|---|---|---|---|---|---|---|
| Baby | 12,637 | 18,646 | 121,651 | 99.95% | 5 | 5 | 663 | 1 |
| Toys Games | 18,748 | 30,420 | 161,653 | 99.97% | 5 | 5 | 1,288 | 19 |
| Sports | 21,400 | 36,224 | 209,944 | 99.97% | 5 | 5 | 2,081 | 18 |

**Table 2: Performance comparison of different recommendation methods over three datasets. The best results are highlighted in bold.**

| Datasets | Baby | | Toys Games | | Sports | |
|---|---|---|---|---|---|---|
| Metrics | Recall | NDCG | Recall | NDCG | Recall | NDCG |
| NeuMF | 0.0502 | 0.0224 | 0.0253 | 0.0128 | 0.0330 | 0.0157 |
| NGCF | 0.0694 | 0.0313 | 0.0970 | 0.0587 | 0.0707 | 0.0337 |
| DGCF | 0.0788 | 0.0465 | 0.1262 | 0.1085 | 0.1026 | 0.0629 |
| AD-DRL$_{ID}$ | 0.0852 | 0.0529 | 0.1517 | 0.1429 | 0.1120 | 0.0722 |
| JRL | 0.0579 | 0.0266 | 0.0472 | 0.0413 | 0.0368 | 0.0214 |
| MMGCN | 0.0814 | 0.0496 | 0.1171 | 0.1065 | 0.0913 | 0.0572 |
| MAML | 0.0867 | 0.0521 | 0.1183 | 0.1117 | 0.1029 | 0.0676 |
| GRCN | 0.0883 | 0.0541 | 0.1336 | 0.1236 | 0.1065 | 0.0693 |
| DMRL | 0.0906 | 0.0561 | 0.1434 | 0.1331 | 0.1111 | 0.0711 |
| BM3 | 0.0911 | 0.0424 | 0.1147 | 0.0683 | 0.1121 | 0.0536 |
| AD-DRL | **0.0968***  | **0.0588*** | **0.1524*** | **0.1435*** | **0.1200*** | **0.0756*** |
| Improv. | 6.84% | 4.81% | 6.28% | 7.81% | 8.01% | 6.33% |

The symbol * denotes that the improvement is significant with $p-value < 0.05$ based on a two-tailed paired t-test.

model parameters are saved every five epochs. Early stopping strategy [34] is performed, *i.e.*, premature stopping if Recall@20 does not increase for 50 successive epochs.

## 4.2 Performance Comparison

We summarize the overall performance comparison results in Table 2. The methods in the first block solely use the user-item interactions, while the methods in the second block also exploit textual and visual information along with the user-item interactions. From this table, we can have the following observations:

- NeuMF, NGCF, DGCF and AD-DRL$_{ID}$ are methods trained exclusively on user-item interactions. Among them, NeuMF outperforms traditional MF-based methods [10], benefiting from deep neural networks' ability to model the non-linear interactions between users and items. NGCF and DGCF achieve state-of-the-art results by utilizing high-order information, with DGCF excelling over NGCF by leveraging disentangled representation to capture diverse user intents, showcasing its effectiveness in enhancing the robustness of user and item representations. Furthermore, although both DGCF and AD-DRL$_{ID}$ employ disentangled representation learning, the performance of AD-DRL$_{ID}$ is significantly better than that of DGCF. This highlights the superiority of our attribute-driven disentangled representation learning approach.

- In general, the multimodal recommendation methods perform better than those only using user-item interactions, demonstrating the effectiveness of leveraging multimodal information on learning user and item representations. Although the simple neural network structure results in lower performance of

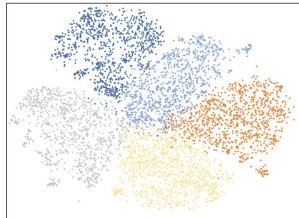
**(a) Disentangled vectors (User ID)**

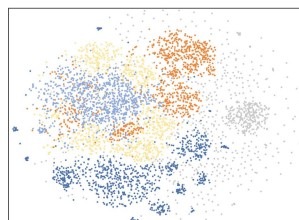
**(b) Disentangled vectors (Item ID)**

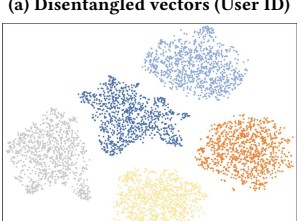
**(c) Disentangled vectors (Review)**

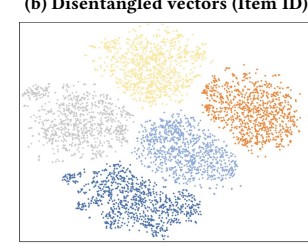
**(d) Disentangled vectors (Image)**

**Figure 2: Visualization of disentangled vectors from ID embeddings and different modalities, with distinct colors indicating different attributes: yellow for *price*, orange for *popularity*, grey for *brand*, dark blue for *category*, and light blue for *others*.**

JRL compared to the graph-based methods (NDCG and DGCF), incorporating rich multi-modal information enables it to outperform NeuMF. By exploiting user-item interactions to guide the representation learning in different modalities, MMGCN yields better performance over NGCF on all the datasets. MAML outperforms NGCF by modeling users' diverse preferences by using multimodal features of items. GRCN surpasses MAML by employing modality features to discover and prune potential false-positive edges on the user-item interaction graph. DMRL captures the different contributions of features from diverse modalities for each disentangled factor, achieving superior results.

- Furthermore, our proposed AD-DRL model consistently outperforms all baselines over all three datasets by a large margin. We credit this to the joint effects of the following two aspects. Firstly, robust user and item representations can be derived using attributes at different levels to guide the disentangled representation learning process. Secondly, incorporating the additional attribute information into representation learning can alleviate the data sparsity problem in the recommendation.

## 4.3 Visualization of Disentangled Representations

AD-DRL conducts disentangled representation learning for user and item representations at two levels of attribute granularity: high

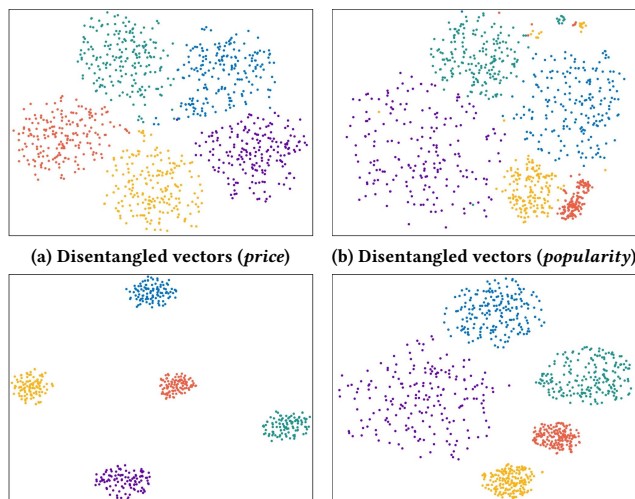

(a) Disentangled vectors (*price*)   (b) Disentangled vectors (*popularity*)

(c) Disentangled vectors (*brand*)   (d) Disentangled vectors (*category*)

**Figure 3: Visualization of the disentangled vectors corresponding to different attributes, where different colors represent different attribute values.**

and low. To further confirm the effectiveness of disentangled representation learning by AD-DRL, we use t-SNE [32] to cluster and visualize the disentangled vectors of users and items from the Sports dataset at each granularity level.

*4.3.1 High-Level Attribute-driven Disentanglement.* Figure 2 displays the disentangled vectors processed by the high-level attribute-driven disentanglement module for each modality feature, where dots of the same color denote the vectors corresponding to the same attribute. It can be observed that our model effectively distinguishes representation vectors corresponding to different attributes for both users and items across various modalities, demonstrating the effectiveness of our model in disentangling at the attribute level. Such disentangled representations can better capture user preferences and item characteristics toward different attributes.

*4.3.2 Low-Level Attribute-driven Disentanglement.* To verify the effectiveness of low-level (attribute value-level) disentanglement, we visualize the representations of each attribute factor (*i.e.*, $v_y^k$ in Equation 9) in Figure 3. The dots of different colors represent different attribute values[3]. As observed, the representations of different attribute values learned from AD-DRL are well separated and the representations of the same attribute value are concentrated. These results demonstrate that AD-DRL can achieve finer-grained disentanglement at the attribute value level, allowing AD-DRL to capture better user preferences.

## 4.4 Interpretability Study

To gain a deeper insight into the interpretability of our model, in this section, we provide some qualitative examples in Figure 4. As an illustrative example, we randomly sample two users (*u*18629

---

[3]For *brand* and *category*, since there are too many attribute values in the dataset, making it difficult to display all of them in Figure 3. Therefore, for these two attributes, we only selected the top 5 attribute values with the most corresponding items in the dataset for demonstration.

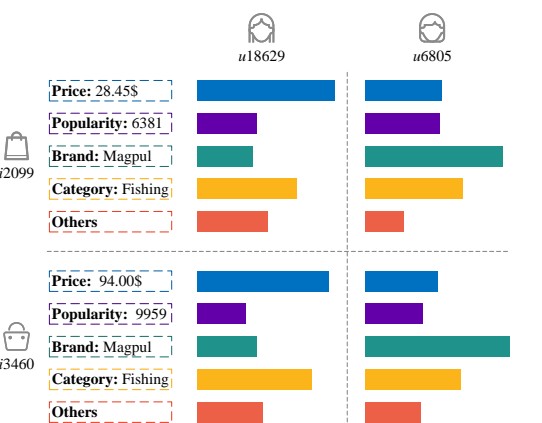

**Figure 4: The preference scores of two users (*u*18629 and *u*6805) for different attributes of two items (*i*2099 and *i*3460).**

and *u*6805) from the Sports dataset who purchased the same items (*i*2099 and *i*3460). Since we have disentangled the representation of users and items based on the attributes, we can analyze how each attribute contributes to recommendation results based on Equation 11. This greatly enhances the interpretability of our model. For example, *price* contributes most to the interaction (*u*10826, *i*2099). This suggests that *u*10826 prefer *i*2099 due to its *price*. However, *u*6805 is more likely to purchase the *i*2099 because of her preference for its *brand*. Such observation indicates that different users have diverse preferences for the same product. In addition, we can also see that user preferences exhibit consistency. For example, *u*6905 values the *brand* of the product more than its *popularity* or *price*, which is different from *u*10826.

## 4.5 Controllability Study

In this section, we evaluate the controllability of AD-DRL by manipulating user preferences for certain attributes. Specifically, we change the user $u$'s preference score for a specific attribute $a$ in Equation 11 to assess if such change yields the anticipated variation in recommendation results. We modify the formula for $s_{u,i}$ in Equation 11 as follows:

$$s_{u,i} = \xi * s_{u,i,a} + \sum_{k \neq a} s_{u,i,k}, \tag{14}$$

where $\xi$ represents the scaling factor for $s_{u,i,a}$, used to adjust the impact of attribute $a$ on $s_{u,i}$. For a specific attribute (in this section, we selected *price* and *popularity* for study), we systematically vary $\xi$ through values of 2, 1, 0.5, 0, and -1. This process allows us to meticulously examine the resultant shifts in our method's recommendation outputs, with findings illustrated in Figure 5.

We conduct a detailed analysis using the *price* attribute as an example. As shown in Figure 5a, we select 100 users from *Baby* dataset who always purchase inexpensive items based on their purchase records. Various colors in Figure 5a denote distinct price levels, showcasing the distribution of items recommended by AD-DRL across these levels for a given value of $\xi$. By comparing the recommendations made by AD-DRL under different values of $\xi$, we have the following observations:

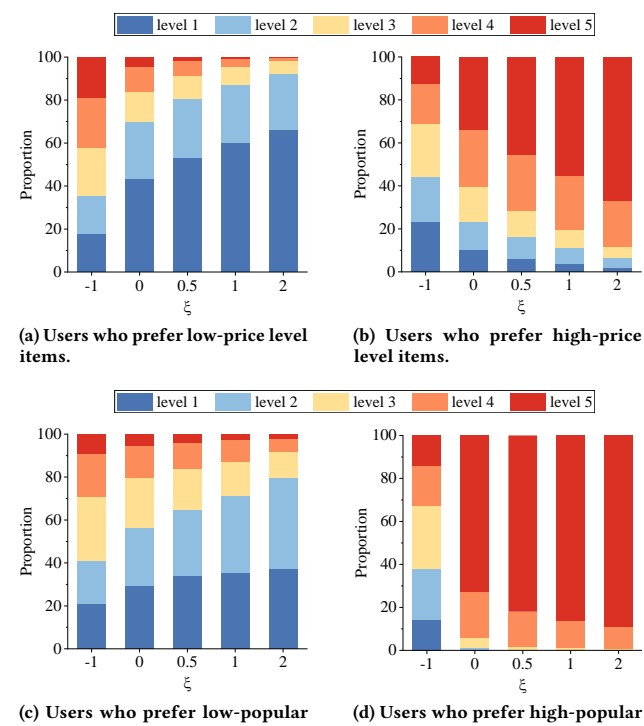

(a) Users who prefer low-price level items.

(b) Users who prefer high-price level items.

(c) Users who prefer low-popular level items.

(d) Users who prefer high-popular level items.

**Figure 5: The proportion of items with different attribute values within the AD-DRL's recommendations when $\xi$ takes different values in Equation 14.**

- When $\xi = 1$, that is, using the original AD-DRL method, it can be seen that AD-DRL is capable of capturing the preference of these users for inexpensive items and recommends more affordable items to them.
- When $\xi = 2$, meaning we have increased the impact of *price* on the user preference score, it can be observed that AD-DRL tends to recommend more inexpensive items to these users.
- When $\xi = 0.5$ or $0$, meaning we reduce or even eliminate the impact of *price* on the user preference score, we find that the recommended items from AD-DRL are more diverse in terms of *price*. It is worth noting that even when $\xi = 0$, the items recommended by AD-DRL still tend to be inexpensive. This could be attributed to the retained chunk associated with the *brand*, and there are correlations between *brand* and *price*. For example, some high-end brands offer products at a premium price, while others provide more affordable options.
- More interestingly, when $\xi = -1$, meaning we have AD-DRL recommend items that are opposite to the users' *price* preferences, it can be observed that AD-DRL indeed recommends some more expensive items.

The above results reveal that adjusting the value of $\xi$ enables us to align AD-DRL's recommendation results with our expectations, thereby illustrating the controllability of AD-DRL. Similar results can be seen in another group of users who prefer high-price level items.

**Table 3: Ablation study of our proposed AD-DRL method over three datasets. The best results are highlighted in bold.**

| Datasets | Baby | | Toys Games | | Sports | |
|---|---|---|---|---|---|---|
| Metrics | Recall | NDCG | Recall | NDCG | Recall | NDCG |
| AD-DRL$_{w/o}$ disentangling | 0.0888 | 0.0550 | 0.1452 | 0.1375 | 0.1152 | 0.0741 |
| AD-DRL$_{w/o}$ intra | 0.0925 | 0.0567 | 0.1492 | 0.1397 | 0.1164 | 0.0750 |
| AD-DRL$_{w/o}$ inter | 0.0959 | 0.0585 | 0.1517 | 0.1429 | 0.1196 | 0.0754 |
| AD-DRL$_{w/o}$ high | 0.0903 | 0.0564 | 0.1492 | 0.1391 | 0.1160 | 0.0748 |
| AD-DRL$_{w/o}$ low | 0.0916 | 0.0553 | 0.1492 | 0.1394 | 0.1177 | 0.0755 |
| AD-DRL | **0.0968** | **0.0588** | **0.1524** | **0.1435** | **0.1200** | **0.0756** |

## 4.6 Ablation Study

To validate the effects of the key components in AD-DRL, we set up the following model variants: 1) AD-DRL$_{w/o}$ disentangling: we do not perform disentangled representation learning during the training process; 2) AD-DRL$_{w/o}$ intra: we only remove the intra-modality disentanglement module; 3) AD-DRL$_{w/o}$ inter: we only remove the inter-modality disentanglement module; 4) AD-DRL$_{w/o}$ high: we remove the high-level attribute-driven disentangled representation learning module; 5) AD-DRL$_{w/o}$ low: we remove the low-level attribute-driven disentangled representation learning module.

Table 3 shows the experimental results for all variants. From the results, we have the following observations:

- Using either high-level or low-level disentangled representation learning independently can significantly improve the performance. This indicates that both disentangled representation learning within modalities and between modalities have a positive effect.
- The AD-DRL combines high-level and low-level disentangled representation learning modules, resulting in significantly improved performance compared to the devised two variants. This demonstrates the necessity and rationality of combining the attribute and attribute value levels.
- Delving deeper into the high-level disentangled representation learning module, removing the intra-modality disentanglement module results in a greater decrease in model performance compared to removing the inter-modality disentanglement module. This indicates that intra-modality disentanglement is the fundamental aspect of disentangled representation learning, while inter-modality disentanglement can further enhance model performance.

## 5 CONCLUSION

In this paper, we highlight the limitations of existing disentangled representation learning techniques in recommender systems. The current methods disentangle the underlying factors behind user-item interactions in an unsupervised manner, leading to limited interpretability and controllability. To overcome this limitation, we propose an attribute-driven disentangled representation learning method, termed AD-DRL, which disentangles attribute factors in various multimodal features. More specifically, the proposed AD-DRL enables disentangling within each modality feature and ensures consistency of representations across modalities at the attribute level. Moreover, it leverages the intrinsic relationships between items sharing the same attribute value. Experimental results demonstrate the superiority of our proposed AD-DRL and showcase its capability in terms of interpretability and controllability.

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
