# OpenReview forum: "Attribute-driven Disentangled Representation Learning for Multimodal Recommendation"
_acmmm.org/ACMMM/2024/Conference — MM2024 Poster_

### Official Review · Reviewer_nxyh · 2024-04-27

**Rating:** 4
**Confidence:** 2

**Summary:**

The paper introduces AD-DRL, a method for improving multimodal recommendation systems by using attribute-specific disentangling of user and item features, which enhances both interpretability and recommendation performance.

**Strengths:**

1. The paper extends existing disentangled representation learning approaches by incorporating both high-level and low-level attribute disentanglements. This dual-level disentanglement is designed to enhance the robustness and independence of feature representations, thereby improving the system's performance.
2. The evaluation of AD-DRL is thorough, with extensive experiments conducted on three real-world datasets to demonstrate its superiority over existing methods.

**Limitations:**

1. In Sec. 3.1, you mention the use of a user-item interaction matrix and multimodal features for learning robust representations. However, how do you ensure that the model does not overfit to specific attributes of user-item interactions, especially in sparse data scenarios?
2. In Sec. 3.2, it's stated that the AD-DRL model disentangles factors at both attribute and attribute-value levels. Could the overlapping of attributes in multimodal data lead to ambiguous or conflicting representations?
3. In Sec. 4, the empirical results show improvements over traditional methods. However, how do you address the variability in results that might stem from different dataset characteristics or external factors affecting the datasets?
4. Given the complexity introduced in the model by handling both high-level and low-level disentanglement, as discussed in Sec. 3.2.2, what is the computational overhead of AD-DRL compared to standard multimodal recommendation systems?
5. In Sec. 4.2, you present a comparison of AD-DRL with several baseline models. How does the performance of AD-DRL scale with increasing data dimensions and what are the implications for larger, more diverse datasets?
6. Considering the attribute-driven approach outlined in Sec. 3.2.1, is there a potential for the model to propagate or amplify biases present in the attribute data, and how might this impact the fairness of the recommendations?

**Suitability:**

3

---

### Official Review · Reviewer_GdX7 · 2024-05-27

**Rating:** 4
**Confidence:** 3

**Summary:**

This paper introduces a novel method, Attribute-Driven Disentangled Representation Learning (AD-DRL), to address challenges in recommendation algorithms that predict user preferences based on historical data. Traditional approaches often struggle to discern the influence of specific factors, like product categories or brands. AD-DRL explicitly incorporates attributes from various modalities into the learning process, allowing for the disentanglement of factors at both attribute and attribute-value levels. This method enhances the interpretability and robustness of the system. Empirical evaluations on three public datasets demonstrate AD-DRL's effectiveness and improved control over recommendation outcomes, proving it a significant advancement in recommendation system technology.

**Strengths:**

1.The paper is well-structured and concise without sacrificing necessary detail, making it easy to follow and understand the key contributions and findings.
2.The proposed AD-DRL, involving multi-level and multi-modal disentanglement, can improve the clarity and usability of disentangled representations in recommendation systems, thereby enhancing both the interpretability and the controllability of the items provided to the user.
3.The descriptions of the datasets and baselines are thoroughly detailed, and the experimental setup is appropriately structured and reasonable.

**Limitations:**

1.Although AD-DRL is able to enhance the performance of existing models, there is no provided code/implementation link to reproduce the results, making it impossible to use as a baseline for future methods.
2.In the Interpretability Study section, the use of merely two examples to demonstrate the advantages of AD-DRL in terms of interpretability is not convincing.
3.In terms of Interpretability and Controllability, the manuscript does not compare the proposed method with baseline models, which is a necessary analysis to establish the efficacy and advancements of the new approach over existing methods.

1.Indicator evaluations or a substantial number of case studies are required for a robust Interpretability Study.
2.It is crucial to include a comparative analysis with established baseline models to validate the claims regarding the interpretability and controllability of the proposed AD-DRL model, presenting both quantitative metrics and qualitative insights to demonstrate its efficacy.

**Suitability:**

2

---

### Official Review · Reviewer_f6Cb · 2024-06-01

**Rating:** 2
**Confidence:** 3

**Summary:**

This paper introduces a novel method called Attribute-Driven Disentangled Representation Learning (AD-DRL), which explicitly incorporates attributes from different modalities into the disentangled representation learning process. Specifically, the article divides the representations of users, items, texts, and images into multiple chunks, each chunk corresponding to a specific item attribute. In each module, this paper uses a self-supervised loss function for alignment. Experiments demonstrate the effectiveness of the proposed method.

**Strengths:**

S1: This paper attempts to add interpretability to the disentangled features, which is an interesting topic.

S2: The organization of the paper is clear and makes it easy to understand.

S3: The paper provides many experimental details to enhance the reproducibility.

**Limitations:**

W1: In Section 3.1, the authors say that the data in this paper includes multimodal features and a user-item interaction matrix, do we need additional information about the item's features such as price and brand?

W2: In the inter-modality disentanglement module, take textual data as an example, if two items have the same attributes, why the textual embedding of these two items is regarded as the negative pair?

W3: In the case of textual information, for example, the embedding of each passage might have a specific meaning. For example, suppose a user evaluates an item “I like this watch because it's a red color” or, “I like this watch because it's a cheap price”. At this point, it doesn't make sense to divide the text embedding into several chunks so that each chuck can predict the attribute. If you do so, maybe the embedding of the "red color" or the "cheap price" will predict the brand. The same thing holds for the image information.

W4: Are the results presented in Figure 4 verifiable? How to conclude that the contributions of each attribute are correct?

W5: How stable is the ranking order for features? For example, does the ordering of “price, popularity, brand” and the ordering of “brand, price, popularity” have any effect on the results? Because the interpretation of embedding will change at this point.

I'd like to modify my rating based on the rebuttal.

**Suitability:**

2

---

### Official Review · Reviewer_aF4X · 2024-06-01

**Rating:** 3
**Confidence:** 3

**Summary:**

This paper introduces a disentangling method, which attempts to make learned representations includes information about the item features explicitly. Specifically, this paper aligns the same information within and between modalities based on the self-supervised loss. In addition, this paper uses attention mechanism to adjust the weights of different modal embedding. Finally, this paper conducts experiments on real-world datasets, and the experimental results demonstrate that the proposed method outperforms existing baseline methods.

**Strengths:**

1. This paper is well organized and makes it easy for the reader to understand the core ideas.

2. The experimental results demonstrate that the proposed method outperforms existing baseline methods.

**Limitations:**

1. If the attributes of two items are very similar, why they need to be treated as the negative pair in the inter-modality disentanglement part?

2. Does it make sense to split embedding into chucks and then align them? For example, if the embedding itself doesn't include information (e.g., an embedding generated from background in an image or a non-solid word in text), does such an embedding have the ability to predict attributes?

3. How do we verify the correspondence between the learned representation and the attributes? In addition, how to ensure that the embedding explicitly learns information about features?

4. Additional sensitivity analysis experiments are needed for $\alpha$, $\beta$, $\gamma$, and a scalar temperature parameter.

**Suitability:**

2

---

### Meta-Review · Area_Chair_atb9 · 2024-06-29

**Recommendation:** Accept (Poster)
**Confidence:** 3

**Metareview:**

This paper is well-organized and achieves high performance. Although one reviewer gives a borderline rejection score, this paper gets overall positive scores and three reviewers raise their scores after the rebuttal phase. To this end, I recommend accepting this paper.